# Enhancing Language Models with Idiomatic Reasoning

**Jianing Zhou**
University of Illinois Urbana-Champaign
Urbana, IL 61801, USA
zjn1746@illinois.edu

**Ziheng Zeng**
University of Illinois Urbana-Champaign
Urbana, IL 61801, USA
zzeng13@illinois.edu

**Hongyu Gong**
Facebook AI
Seattle, WA 98109, USA
hygong@fb.com

**Suma Bhat**
University of Illinois Urbana-Champaign
Urbana, IL 61801, USA
spbhat2@illinois.edu

## Abstract

Advancements in Large Language Models (LLMs) have significantly propelled the field of Natural Language Processing (NLP); however, nuanced reasoning in the presence of non-canonical language forms, such as figurative language, remains an intricate challenge. These language forms, integral to human communication, elude standard LLM comprehension due to their inherent non-compositionality, contextual ambiguity, and sparse representation in text corpora. Addressing these challenges, this paper introduces an innovative approach to seamlessly incorporate idiomatic knowledge into pre-trained language models (PTLMs). Our methodology first employs a novel *multi-view* data augmentation strategy that uses idiomatic instances representing one property to generate training data for various idiom-related tasks. When combined with a novel parameter-efficient tuning mechanism that accounts for the unique attributes of idiomatic language, we embed task-specific and idiomaticity-aware inductive biases within a PTLM. Integrating a meta-pretraining protocol based on meta-learning principles, further equips the model with enhanced adaptability to diverse downstream idiom-aware tasks. Empirical validation on diverse benchmarks centered around idiom comprehension and reasoning, demonstrates the efficacy of our approach. Notably, our model surpasses various parameter-efficient fine-tuning baselines outperforming the conventional full fine-tuning paradigms, thereby creating more contextually aware and linguistically robust language models.

## 1 Introduction

*Idiomatic expressions* (IEs), commonly found in natural language, encompass a broad range of dynamic figures of speech that enhance fluency and concision across various genres and play a key role in human communication (Moon et al., 1998; Baldwin & Kim, 2010a). However, pre-trained language models (PTLMs), such as BART (Lewis et al., 2020), T5-large (Raffel et al., 2020), and T0, struggle to reason in the presence of IEs (Chakrabarty et al., 2021b), failing to understand their figurative semantics (Bhargava & Ng, 2022; Zeng & Bhat, 2022). Despite their impressive scale of pre-training, even large language models (LLMs) such as GPT-4 and LlaMa2 struggle to reason effectively in the presence of figurative expressions (Liu et al., 2023; Kabra et al., 2023). This is in part due to IEs' long tail distribution in natural language, their *non-compositionality*, i.e., the overall meaning of an IE is not directly derived from its individual components (Sag et al., 2002), and their *contextual ambiguity*, i.e., an IE can be interpreted idiomatically or literally depending on the context (Baldwin & Kim, 2010b). Beyond this, it is known that comprehension of idioms requires a broad linguistic awareness of the relationship between meaning and surface form, apart from commonsense

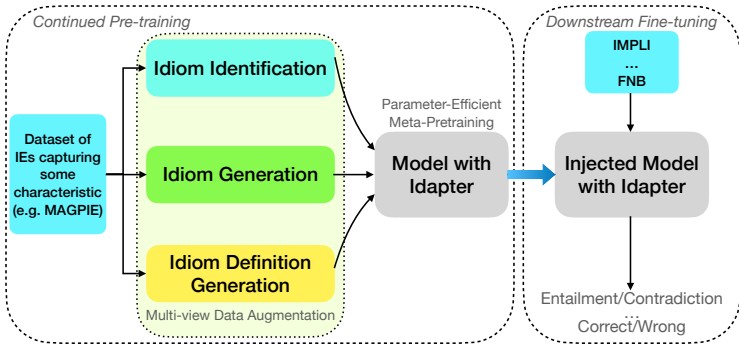

Figure 1: The overview of our framework.

associated with their use (Cacciari & Levorato, 1989). Therefore, it is necessary to provide idiomatic knowledge to the PTLMs for better reasoning in the presence of idioms.

Recent works (Škvorc et al., 2022; Zeng & Bhat, 2021) have enhanced PTLMs for specific IE tasks, such as IE identification and usage recognition, without aiming for generalizable IE comprehension ability. Others (Zeng & Bhat, 2022; 2023) adapt PTLMs to learn semantically meaningful IE representation, thus rendering PTLMs suitable for multiple downstream IE tasks with limited reasoning abilities. A recurring obstacle encountered by previous studies is incorporating IE-related knowledge when IE-related data is scarce: MAGPIE (Haagsma et al., 2020), the largest and the most popular IE corpus, contains 56k sentences covering 1755 IEs. As we will show in this study, scaling up pre-trained models with more parameters and larger pre-training data than ever before fails to automatically resolve this deficiency. While open-source LLMs also lack IE reasoning ability, in this study, we show how state-of-the-art IE comprehension and reasoning abilities are achievable by explicitly tackling the data scarcity challenge. Remarkably, we can accomplish this without relying on astronomically expensive pretraining data or parameter up-scaling; instead we adapt smaller-frame PTLMs to various downstream tasks through meta-learning via novel data augmentation and parameter-efficient tuning methods.

Figure 1 summarizes our approach. Given a dataset of sentences with IEs, we propose *multi-view data augmentation* where a single dataset is transformed into an array of instances for multiple tasks, each representing an IE-related characteristic, e.g. identification task for learning contextual ambiguity, definition generation task for learning non-compositionality and generation cloze task for learning the usages. Then, we continued pre-train a PTLM on these tasks to learn generic IE knowledge. Unlike traditional data augmentation methods that generate more data for the same task or generate different data for different tasks that are irrelevant, we use the *same* data but transform them into *different* tasks. Sharing the same IEs and the different forms of sentences (as different tasks), permits the model to learn IE-related properties introduced by each task *and* the shared IE knowledge across tasks.

To address the training efficiency challenge, we utilize parameter-efficient fine-tuning (PEFT) during continued pre-training and fine-tuning for downstream tasks. Traditional PEFT methods (Rebuffi et al., 2017; Guo et al., 2021; Hu et al.; Pfeiffer et al., 2020a) are primarily designed to learn from single-task data while ignoring linguistic processes (e.g., non-compositional language). In contrast, our data augmentation leads to mixed-task training batches containing exclusively non-compositional language. Therefore, inspired by Adapters (Pfeiffer et al., 2020b), we propose the *Idapter*. It enables appropriate token representation by generating a token-level non-compositionality scalar to linearly combine the compositional and non-compositional representations in accordance with the its contextual semantics. Further, it enables a mixed-task batch training by generating a task-specific scalar to alter the intermediate representations for each task. Finally, towards orienting the model to different downstream tasks, we also propose a meta-learning based *meta-pretraining* mechanism that optimally initializes the model parameters for effective downstream use.

To showcase the efficacy of our method, we first continue pre-training the Idapter attached to a frozen backbone PTLM using our multi-view augmented data and meta-pretraining mechanism. Then, we deploy our IE-knowledge-enhanced model on different downstream

IE-related tasks (see Figure 1). Empirical results show our method outperforms not only other LMs that are fine-tuned fully or in a parameter-efficient manner but also open-source LLMs such as LlaMa2 and Mistral that are 200 times larger than the models we study.

The contributions of this work are as follows.
(1) We propose a framework that equips PTLMs with idiomatic reasoning, which could then be used to solve different downstream IE-aware tasks;
(2) We address IE data scarcity via a multi-view data augmentation strategy that transforms a given dataset into a set of datasets for continued pre-training;
(3) To inject idiomatic intelligence efficiently, we propose Idapter—an idiom-aware PEFT method designed for continued pre-training with multi-view data augmentation;
(4) To adapt the model to different IE-aware tasks effectively, we propose our meta-pretraining mechanism based on meta-learning;
(5) We show the efficacy of our method on a diverse set of downstream IE-related tasks. Our IE-knowledge enhanced model achieves new state-of-the-art results on IE-related tasks.

## 2 Related Prior Work

**Knowledge Injection.**    Recent studies have shown the promise of integrating *explicit* external knowledge for transformative impacts on downstream NLP tasks by continued pre-training (Zhang et al., 2019b; Beltagy et al., 2019; Lee et al., 2020; Zhang et al., 2022; Yu et al., 2022). Among them, some works explore to inject idiomatic knowledge into PTLMs via task-specific data augmentation (e.g., generating sentence pairs for IE paraphrasing (Zhou et al., 2021c)), leveraging external knowledge (Zeng & Bhat, 2022), and learning with parameter efficient methods (Zeng & Bhat, 2023) (e.g., Adapters (Pfeiffer et al., 2020a)). However, prior works have limited utility in that they only focus on a single task (e.g., identifying IEs (Flor & Klebanov, 2018; Amin et al., 2021; Zeng & Bhat, 2021), disambiguating between their figurative/literal use (Liu & Hwa, 2017; 2018) or paraphrasing (Zhou et al., 2021a;b)). Although Zeng & Bhat (2023) show their model's performance on different downstream tasks, their method is still limited since they utilize adapters without accommodating IE-specific characteristics. Besides, their training method does not mitigate the discrepancy between the training objectives and downstream tasks, rendering their method less adaptable to tackle practical downstream tasks. Additionally, none of the prior works show performance improvements in natural language understanding tasks in the presence of IEs; their performance improvements focus solely on IE processing tasks.

**Data Augmentation.** Data augmentation, generating synthetic task-specific data, is a common, well-studied approach in the NLP community to mitigate data scarcity issues (Feng et al., 2021). The augmented data can be derived from labeled data and directly utilized in supervised learning (Wei & Zou, 2019) or in semi-supervised learning for unlabeled data (Xie et al., 2020). While several approaches have been proposed to address learning with limited labeled data, such as unsupervised pre-training (Devlin et al., 2019; Raffel et al., 2020), multi-task learning (Liu et al., 2017; Augenstein et al., 2018) and semi-supervised learning (Miyato et al.; Xie et al., 2020), these methods have limitations in IEs' domain due to IEs' non-compositionality, which complicates the generation of new texts. Consequently, we introduce a novel multi-view data augmentation technique that circumvents the need to generate new texts with IEs by transforming the original texts into different tasks.

**Parameter-Efficient Fine-tuning.** While conventional fine-tuning updates all parameters of PTLMs, recent research has shown that updating or incorporating a minute fraction (e.g., 0.01%) of the parameters can achieve a performance comparable to that of full fine-tuning. Beginning with Adapters (Rebuffi et al., 2017), a variety of advanced Parameter Efficient Fine-Tuning (PEFT) strategies have been developed, including selecting a sparse subset of parameters for training (Guo et al., 2021), low-rank updates (Hu et al.), implementing optimization in a lesser-dimensional subspace (Aghajanyan et al., 2021), incorporate low-rank Adapters using hypercomplex multiplication (Karimi Mahabadi et al., 2021). However, all prior PEFT strategies are designed for the more generic, compositional expressions, without any special consideration for non-compositional figures of speech such as IEs. More critically, we hypothesize that considering the figurative meanings and degrees of

| **Idiom Identification** |
| --- |
| *Input:* What is the idiomatic expression in the sentence: It looks like the temperature is going to drop tonight , so be careful not to **catch a cold** . |
| *Output:* catch a cold |
| **Idiom Generation** |
| *Input:* Fill in the mask with an idiomatic expression in the sentence: It looks like the temperature is going to drop tonight , so be careful not to [MASK] . |
| *Output:* catch a cold |
| **Idiom Definition Generation** |
| *Input:* What is the definition of the idiom 'catch a cold' in the sentence: It looks like the temperature is going to drop tonight , so be careful not to **catch a cold** . |
| *Output:* become infected with a cold |

Table 1: Examples of input sentences and output sentences in our tasks. Non-compositional expressions are highlighted in **bold red**

non-compositionality of IEs to be unique to each IE, learning each IE can be considered a separate subtask. This renders prior PEFT methods impractical for mixed-task training batches produced by our multi-view data augmentation since these methods require isolated modules for distinct subtasks. Thus, we propose Idapter with non-compositionality and task-specific scaling to aid idiomatic knowledge injection.

**Meta-learning.** The concept of meta-learning, also known as "learning to learn," has garnered substantial attention, resulting in a vast body of literature. Among the various meta-learning frameworks, Model-Agnostic Meta-Learning (MAML) stands out Finn et al. (2017). Notably, MAML-related techniques have found applications across a range of natural language processing (NLP) tasks (Dou et al., 2019; Kang et al., 2020; Qian et al., 2021; Huang et al., 2023). Despite these advancements, a critical gap remains: the application of meta-learning to pre-training of language models. While some NLP researchers have explored meta-learning (Ke et al., 2021; Wu et al., 2022), few have delved into its potential for domain-specific knowledge injection. Our work aims to address this gap by investigating how meta-learning can enhance the pre-training of domain-specific language models, filling a niche that has yet to be fully explored in the literature.

## 3 Multi-View Data Augmentation

Traditionally, PTLMs rely on large-scale corpora as implicit sources of domain-specific knowledge (Lee et al., 2020). However, the long tail nature of idioms and the unavailability of comparable sources focusing on idiomatic expressions—the largest-to-date collection of sentences with IEs, MAGPIE, is made of only 56k sentences of non-uniform quality of context spanning 1,755 IEs—with sufficient contexts from which to learn their meanings, calls for novel approaches to idiomatic knowledge injection.

We propose a multi-view data augmentation technique in which an available dataset of idiomatic sentences, such as MAGPIE, is augmented via different tasks derived from this dataset, to provide a diverse range of idiomatic knowledge. Specifically, given the MAGPIE sentences as input, we create datasets for different tasks by tailoring the corresponding outputs to these ends. To inform the model of IE non-compositionality and the associated figurative meaning, we create the *definition generation* task, in which the model is asked to produce the dictionary definition of the IE from the input. To impart IEs' contextual ambiguity, we introduce the *idiom identification* task in which the model is expected to extract the IEs that are used in their idiomatic sense (e.g., *People talk about Alice behind her back.*), while ignoring IEs that are used in their literal sense (e.g., *Alice put her bag behind her back.*), outputting an empty string. Finally, to allow the model to recognize proper IE uses in a given context, we add the *idiom generation* cloze task in which, given the contextual cues, the model is expected to fill in a mask using the IE with an appropriate figurative meaning. We resort to these tasks because, despite the pretraining objective of using the context to

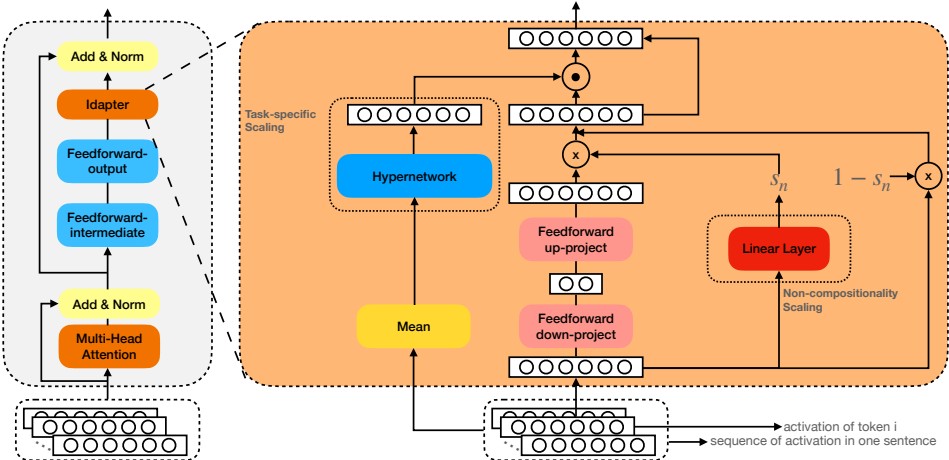

Figure 2: The overview of Idapter. For each token, non-compositionality scaling is used for figurative semantics and task-specific scaling is used for training with mixed-task batches.

predict a given token/word, the emerging understanding is that this is insufficient to learn to represent idioms and much less to reason in their presence (Miletić & Walde, 2024).

As a result, for each sentence in the dataset that uses the IE idiomatically, we create three expected outputs corresponding to these three tasks. A sentence where the IE is used literally gets used only for the idiom identification task.

As an illustration, the input/output pairs of these three tasks for an input sentence with the IE *catch a cold* are shown in Table 1. Since all outputs are generated from the same original IE sentence, we hypothesize that by learning these tasks jointly, the model effectively learns comprehensive knowledge about the (common) IE, *catch a cold*, shared across tasks. In contrast to the traditional data augmentation methods that aim to generate more data for a single task and the traditional multitask learning methods that utilize different data for different tasks, our method creates different tasks from the same data, thus allowing a more effective data utilization in a low-resource scenario.

## 4   Parameter-efficient Training for Idiomatic Knowledge

Advancements in language models have significantly scaled up the number of parameters and the pretraining corpus. However, injecting new knowledge into large PTLMs using PEFT methods, rather than full fine-tuning, makes for a more efficient model. In this paper, we present an improved PEFT method that, instead of being task/process agnostic, incorporates considerations for non-compositional language, thus enabling the training of multiple tasks simultaneously. Specifically, building upon the Adapter framework, we propose an **Idapter**, a novel PEFT method designed to inject idiomatic knowledge into PTLMs using the multi-view augmented data.

**Adapters.** An Adapter is a lightweight neural network placed between or inside the intermediate transformer layers of a PTLM. During training, only the Adapter weights are trainable, while the PTLM's parameters are frozen to improve efficiency and reduce training costs. Similarly, we use the paradigm where the Adapters are placed inside the intermediate layers. With our focus on idiomatic knowledge injection, the Adapter is expected to learn a different representation $\mathbf{h}_i^a$ for each token $i$ to incorporate the potential non-compositional meaning given the original representation $\mathbf{h}_i \in \mathbb{R}^d$ from the frozen PTLM:

$$\mathbf{h}_i^a = \text{Adapter}(\mathbf{h}_i)$$

**Non-compositionality Scaling.** The Idapter takes into account the non-compositionality in its architecture. It has a non-compositionality scaling that generates a non-compositionality

score $s_n^i$ for token $i$:

$$s_n^i = \mathbf{W}_n \mathbf{h}_i + b_n$$

where $\mathbf{W}_n \in \mathbb{R}^{1 \times d}$ and $b_n$ are the trainable parameters of the non-compositionality scaling layer ($n$ stands for "non-compositionality"). Then, for each token, the final representation is the linear combination of the Adapter representation, $\mathbf{h}_i^a$, scaled by the non-compositionality score, $s_n$, and the original PTLM representation scaled by $1 - s_n$:

$$\hat{\mathbf{h}}_i = (1 - s_n^i)\mathbf{h}_i + s_n^i \mathbf{h}_i^a$$

where $i$ refers to token i.

We include the idiom identification task in our continued pre-training to teach the model about idioms' contextual ambiguity and thus disambiguate each token as part of a compositional or non-compositional phrase. Therefore, learning the idiom identification task optimizes the parameters of the non-compositionality scaling layer (referred to as NC layer).

**Task-specific Scaling.** Because our data augmentation method formats data into different tasks, mixed-task batches are used during training, which is a second point of departure from traditional PEFT methods. The Idapter aids mixed-task batch training by learning a scalar vector for each task to help the model distinguish and optimize each task separately. Each task-specific vector is generated by a dedicated, task-specific hypernetwork used only for the inputs from its corresponding task. We consider simple, efficient linear layers as hypernetworks that are functions of the original PTLM's representations, $\mathbf{h}_t \in \mathbb{R}^d$, where $t$ denotes a specific task in our knowledge injection phase. We implement these hypernetworks after each transformer layer from the PTLM. The hypernetwork $\mathcal{H}(\cdot)$ that generates the task-specific vector $l_t \in \mathbb{R}^d$ is defined as:

$$l_t = \mathcal{H}(\mathbf{h}_t) = \mathbf{W}^{\mathcal{H}} \mathbf{h}_t$$

where $\mathbf{W}^{\mathcal{H}} \in \mathbb{R}^{d \times d}$ are the trainable parameters of the hypernetwork. The task-specific vectors are then multiplied by the sequence of representations obtained after the non-compositionality scaling as:

$$\mathbf{h}_i^* = l_t \odot \hat{\mathbf{h}}_i + \hat{\mathbf{h}}_i$$

where $\odot$ represents element-wise multiplication.

In effect, inputs from different tasks will be represented differently by Idapter using task-specific scaling (referred as TS layer), making learning mixed-task batches possible.

## 5 Meta-Pretraining

We argue that most PTLMs' objective is to maximize their performance on pretraining tasks without considering any aspect of the downstream tasks, which results in a discrepancy between pre-trained models and downstream tasks. To alleviate this, we leverage a meta-learning algorithm for pre-training optimization of the Idapter. With different augmented tasks as meta-tasks, we utilize meta-learning to update the initial parameters of our model, which could anticipate to improve downstream performance on tasks involving idiomatic expressions. Our meta-pretraining can be divided into two stages:

i) meta-train stage, which updates the model's parameters $\theta$ using the train set of pre-train data:

$$\theta_k = \theta_{k-1} - \alpha \nabla_{\theta_{k-1}} L\left(\theta_{k-1}; D_{T,k}^{train}\right)$$

ii) meta-test stage, which updates the model's initial parameters $\theta_0$ by one gradient descent step over the test set of pre-train data:

$$\theta_0' = \theta_0 - \beta \nabla_{\theta_0} L\left(\theta_k; D_T^{test}\right)$$

| Method | IMPLI | | | | I2L | | L2I | | FNB | | |
|---|---|---|---|---|---|---|---|---|---|---|---|
| | E | NE | ANT | Acc | B | R | B | R | CC | ICC | Acc |
| Full | 85.21 | 59.71 | 89.96 | 81.16 | 82.46 | 77.15 | 82.89 | 76.69 | 82.82 | 78.03 | 80.42 |
| Adapter | 81.4 | 66.01 | 87.43 | 79.98 | 81.21 | 75.36 | 82.72 | 76.66 | 83.94 | 74.93 | 79.44 |
| Lora | 91.08 | 71.15 | 70.59 | 80.07 | 81.12 | 75.28 | 81.85 | 75.74 | 72.11 | 77.75 | 74.93 |
| $(IA)^3$ | 86.72 | 38.74 | 80.75 | 74.26 | 76.44 | 69.82 | 76.38 | 70.19 | 76.06 | 77.46 | 76.76 |
| PIER+ | 89.39 | 53.54 | 13.07 | 56.78 | - | - | - | - | 66.80 | 63.36 | 65.08 |
| Llama2-10shot | 53.32 | 84.19 | 89.84 | 71.92 | 61.09 | 56.36 | 65.38 | 60.63 | 62.82 | 81.41 | 72.11 |
| Mistral-10shot | 36.05 | **84.58** | **90.11** | 64.21 | 77.12 | 69.24 | 77.88 | 70.55 | 74.08 | 59.72 | 66.90 |
| Ours | 87.86 | 84.19 | 88.77 | **87.34** | **86.11** | **81.32** | **85.78** | **80.79** | 83.10 | **84.23** | **83.66** |
| - w/o Augmentation | 87.02 | 66.32 | 88.47 | 82.95 | 83.31 | 77.69 | 83.05 | 77.16 | 76.62 | 83.10 | 79.86 |
| - w/o Idapter | **94.31** | 77.87 | 75.94 | 84.75 | 83.68 | 78.22 | 83.13 | 77.94 | 76.62 | 83.10 | 80.14 |
|   - w/o NC layer | 93.55 | 78.66 | 78.88 | 85.53 | 84.76 | 79.53 | 84.32 | 79.14 | 82.29 | 79.19 | 80.74 |
|   - w/o TS layer | 94.29 | 79.37 | 79.84 | 86.34 | 85.92 | 80.06 | 85.36 | 79.53 | 81.13 | 81.13 | 81.13 |
| - w/o Meta-Pretrain | 85.23 | 82.21 | 87.61 | 85.34 | 84.45 | 78.69 | 84.33 | 79.21 | 82.65 | 78.99 | 80.82 |

Table 2: Performance of different methods for different downstream tasks. **E**, **NE** and **ANT** refers to the accuracy on the entailment examples, non-entailment examples and antonym non-entailment examples for IMPLI. **B** and **R** refers to the BLEU score and Rouge-2 score respectively. **CC** and **ICC** refers to the accuracy on the correct labels and wrong labels for FNB. **Acc** denotes the overall accuracy averaged based on all the examples. **Bolded** results represent the best performance and underlined results represent second best. The differences on overall accuracies, BLEU scores and Rouge-2 scores between our model and the other baselines are statistically significant at level 0.05.

where $\alpha$ refers to the learning rate during the meta-train stage and $\beta$ refers to the learning rate during the meta-test stage. $D_{T,k}^{train}$ represents the kth batch in train set of pre-train data from task T. $D_T^{test}$ represents the test set of pre-train data from task T.

The returned meta-parameters $\theta_0$ are used as the pre-trained model parameters for different downstream tasks. We utilize Reptile (Nichol et al., 2018), a variant of MAML (Finn et al., 2017) in our meta-pretraining stage for better efficiency.

# 6 Experiments

**Continued Pre-training Tasks.** The continued pre-training tasks aim to inject fundational IE knowledge into the models. We use the MAGPIE (Haagsma et al., 2020) for our multi-view data augmentation. The original dataset was intended for *idiom identification*, where the task is to decide if an IE is used figuratively or literally in a given context. Thus, it serves as the data foundation for the first task of idiom identification. For the task of *idiom definition generation*, we retrieve the definitions from online resources, such as Wiktionary and Google Dictionary. For the task of *idiom generation*, we replace the whole IE with a single [MASK] token. The model is expected to reconstruct the original IE to replace the [MASK] token. By construction and the nature of the tasks, these tasks are learned via self-supervision (idiom generation) and provided labels (definition generation and idiom identification).

**Downstream Tasks.** We further verify the effectiveness of our method on a diverse set of downstream tasks including IE-related natural language inference (IMPLI) (Stowe et al., 2022), Idiomatic Expression Paraphrasing (I2L) (Zhou et al., 2021c), Literal Expression Stylizing (L2I) (Zhou et al., 2021c) and Figurative Narrative Benchmark (FNB) (Chakrabarty et al., 2021a). The tasks differ from the continued pre-training tasks described above and involve a relatively deeper interpretation of idioms. Details are provided in the appendix.

**Experimental Procedure.** Our experiments are performed as follows: We use the T5-large model (Raffel et al., 2020) as the backbone PTLM since its multi-task pre-training aligns with our continued pre-training and thus provides better and faster adaptation. We first add an Idapter to the backbone model and then update only the Idapter during the meta-pretraining stage. After meta-pretraining, we deploy the model on different downstream tasks for fine-tuning.

| Dataset | IMPLI | | | | I2L | | L2I | | FNB | | |
|---|---|---|---|---|---|---|---|---|---|---|---|
| | E | NE | ANT | Acc | B | R | B | R | CC | ICC | Acc |
| 0-task | 87.02 | 66.32 | 88.47 | 82.95 | 83.31 | 77.69 | 83.05 | 77.16 | 76.62 | 83.10 | 79.86 |
| No GEN | 94.12 | 77.08 | 81.28 | 86.22 | 85.06 | 81.21 | 85.69 | 80.44 | 82.28 | 81.15 | 81.72 |
| No IDT | 94.12 | 76.68 | 81.55 | 86.22 | 85.92 | 81.06 | 85.48 | 80.39 | 82.82 | 81.41 | 82.11 |
| No DEF | 93.74 | 78.26 | 79.14 | 85.62 | 84.79 | 80.88 | 85.02 | 80.09 | 81.41 | 80.85 | 81.13 |
| All | **87.86** | **84.19** | **88.77** | **87.34** | **86.11** | **81.32** | **85.78** | **80.79** | **83.10** | **84.23** | **83.66** |

Table 3: A comparison of model performance while varying the number of injection tasks used during pre-training

**Baselines.** To validate our method, we compare it to full fine-tuning and three SOTA PEFT methods including Adapters (Rebuffi et al., 2017), Lora (Hu et al.) and $(IA)^3$ (Liu et al., 2022). We also compare our model with PIER+ (Zeng & Bhat, 2023). To study the extent to which scaling and strategic fine-tuning methods enable idiomatic reasoning, we also compare our proposed method with open-source LLMs, including Llama2-13b (Touvron et al., 2023) and Mistral-7b (Jiang et al., 2023) with 10 in-context examples (the number 10 was empirically found to yield reasonable results beyond which the performance did not improve significantly).

# 7  Results

**IMPLI.** Prior works (Sag et al., 2002; Tayyar Madabushi et al., 2021; Stowe et al., 2022) have demonstrated that the presence of IEs negatively impacts NLI performance. We use this as a case study to demonstrate how our proposed IE-related knowledge injection increases robustness in the presence of IEs. As shown in Table 2, our method efficiently injects idiomatic knowledge into the model. For the models without a PEFT module (Full), our proposed model outperforms the original model by 6.18% in accuracy. Notably, there is a gain of 24.48% accuracy in the non-entailment (NE) class. We point out that a good non-entailment performance is much harder to attain compared to entailment since the model resorts to predicting all instances as entailment as discussed in (Stowe et al., 2022). We ascribe this performance gain to the model's understanding of the IE semantics rather than relying on spurious features towards the entailment predictions, which is further verified by our ablation study in Section 8.

**Paraphrasing & Stylizing.** Here we present the results of two related tasks: I2L and L2I. As presented in Tables 2, our proposed approach substantially benefits these two tasks by incorporating idiomatic knowledge. Similarly, our model achieves the highest performance. It surpasses the model trained solely on the I2L dataset by a significant margin of 3.65 points in BLEU score (82.46 vs. 86.11) and outperforms the model trained solely on the L2I dataset by a margin of 2.89 points in BLEU score (82.89 vs. 85.78). For both tasks, this improvement holds consistently across models with different PEFT modules, showcasing the effectiveness of our method. Considering that BLEU and ROUGE scores only focus on the n-gram overlapping, we also utilize Bertscore to evaluate our proposed method. It is shown that our model also achieves the best performance on Bertscore. The details are provided in Table 6 in the appendix.

**FNB.** As shown in Table 2, our method efficiently equips the model with the ability of reasoning in the presence of idioms. For the models without a PEFT module (Full), the model trained on our multi-view augmented data (+PT) outperforms the original model by 3.24% in accuracy.

We note that the tasks we study (IMPLI, L2I, I2L, and FNB) contain IEs not covered by idiomatic knowledge injection, further demonstrating our model's generalizability to IEs unseen before fine-tuning. Details are provided in the Section **??** in the appendix.

| Method | IMPLI | | | | I2L | | L2I | | FNB | | |
|---|---|---|---|---|---|---|---|---|---|---|---|
| | E | NE | ANT | Acc | B | R | B | R | CC | ICC | Acc |
| Full | 78.75 | 68.38 | 86.1 | 78.86 | 78.9 | 73.31 | 81.22 | 74.21 | 13.80 | 86.48 | 50.14 |
| Adapter | 75.33 | 60.24 | 82.01 | 74.19 | 79.25 | 72.74 | 80.03 | 73.36 | 19.15 | 82.82 | 50.99 |
| Lora | 70.21 | **79.05** | **92.78** | 79.46 | 78.11 | 71.36 | 79.05 | 72.4 | 40.56 | 62.25 | 51.41 |
| $(IA)^3$ | 80.22 | 62.38 | 78.13 | 75.63 | 72.95 | 68.66 | 73.04 | 68.6 | 19.44 | 81.41 | 50.42 |
| Ours | **93.55** | 77.47 | 81.82 | **86.22** | **82.82** | **76.92** | **83.01** | **77.81** | **82.54** | 74.65 | **78.59** |

Table 4: Performance of different methods for different downstream tasks when only 20% of data is used for downstream fine-tuning.

## 8 Analysis

We gain more insights about our multi-view data augmentation method and Idapter via further analyses.

**Ablation Study.** In our ablation studies, we analyze the impact of excluding different components from our proposed model. Table 2 demonstrates that omitting any part of our model results in decreased performance, highlighting the effectiveness of our approach. Specifically, when the multi-view data augmentation is excluded, performance is significantly worse, emphasizing the importance of this augmentation method. Additionally, excluding either the non-compositionality scaling layer or the task-specific scaling layer leads to performance degradation, reinforcing the effectiveness of the various inductive biases incorporated in our proposed Idapter.

**Injection Task Influence.** To study the influence of different continued pre-training tasks, we pre-train the Idapter with the absence of different tasks. For the 0-task case, we directly fine-tune the Idapter with the backbone PTLM on the downstream tasks with no pre-training injection tasks. Then we exclude different injection tasks from the pre-training data and pre-train the Idapter with the remaining two tasks. Finally, for the 3-task case, we first pre-train the Idapter on all three augmented tasks, adding IE appropriateness compared to the 2-task case. After the pre-training with augmented tasks, we further fine-tuned the Idapter on different downstream tasks. As shown by the results in Table 3, all instances of excluding different injection tasks degrade the performance across all four downstream tasks, while pre-training with all three tasks provides the best performance. It should be noted that excluding the definition generation task impacts the performance the most, presumably because this task directly injects the meaning of idioms into the model.

**Data efficiency.** Another question concerns whether a model equipped with generalized idiomatic knowledge is able to learn different downstream tasks in a data-efficient way. To this end, we fine-tune the continued pre-trained Idapter on the downstream tasks with fewer data. We use only 20% of the original data in different downstream tasks for fine-tuning and summarize the results in the Table 4. As is shown, using only 20% of the data, our proposed model achieve the best performance. In fact, the model augmented with the Idapter achieves a competitive performance that is only slightly worse than the model fully fine-tuned on the entire data for IMPLI; this confirms the data efficiency of our method.

**Zero-Shot Setting For IMPLI.** Our multi-view data augmentation method shows an improvement in a zero-shot, out-of-distribution setting. We observe this by fine-tuning the full model only on the MNLI dataset (Williams et al., 2018), a generic NLI dataset with a negligible number of IE instances, and test on IMPLI; models trained with the multi-view augmentation data show an improved overall performance on the IMPLI dataset (+8.04% gain in accuracy). Our method's notable improvement comes from a 1.24% accuracy increase in NE class, 2.1% in ANT class and 7.43% in the E class. Details are provided in Table 7 in the appendix.

**Qualitative Analyses.** Some idioms may only appear in the downstream tasks, but not in our continued pre-training data. Therefore, we explore our model's performance on idioms seen during pretraining and those unseen for different downstream tasks. As shown in

Table 5, across different downstream tasks, our model was able to process unseen idioms, though not nearly as well as those seen.

Beyond processing idioms, we find that our model is able to positively impact the processing of other types of figurative expressions, including metaphor detection (+0.8% on accuracy) and euphemism detection (+1.03% on accuracy). Details are provided in appendix A.4.

| Method | IMPLI | | | | | I2L | | | L2I | | | FNB | | | |
|--------|-------|-------|-------|-------|-----|-------|-------|------|-------|-------|------|-------|-------|-------|-----|
| | E | NE | ANT | Acc | Num | B | R | Num | B | R | Num | CC | ICC | Acc | Num |
| Seen | 94.39 | 76.57 | 81.41 | 87.37 | 336 | 90.67 | 86.90 | 551 | 93.30 | 89.35 | 551 | 83.33 | 84.33 | 84.67 | 320 |
| Unseen | 89.66 | 80.00 | 85.00 | 85.94 | 18 | 80.69 | 75.70 | 618 | 77.01 | 73.09 | 618 | 85.45 | 70.91 | 78.18 | 72 |
| All | 87.86 | 84.19 | 88.77 | **87.34** | 354 | **86.11** | **81.32** | 1169 | **85.78** | **80.79** | 1169 | **83.10** | **84.23** | **83.66** | 392 |

Table 5: Performance on seen idioms and unseen idioms. **Num** refers to the number of idioms in the test set.

# 9 Conclusion and Future Works

In this work, we propose a continued pre-training method based on a novel multi-view data augmentation approach, a parameter-efficient training method, Idapter and a meta-pretraining mechanism. Unlike previous works on IEs that only focus on one task, we empirically show that our method injects generic idiomatic knowledge that is beneficial for various tasks involving IEs, such as IE-related natural language inference, idiomatic expression paraphrasing, literal expression stylizing and figurative narrative reasoning. Besides, we also show that even with only 20% of the training data for downstream IE-related tasks, the model equipped with generalized idiomatic knowledge yields comparable performance with the model trained on the whole data on IE-related natural language inference. Future work should explore how this method can be useful in low-resource scenarios, such as the medical and legal domains.

## Limitations

The main limitation of our method is that it still needs external resources to obtain the definitions of IEs, which restricts the learning of some IEs whose definitions cannot be easily accessed.

Also, the scope of this research focuses on equipping large PTLMs with idiomatic knowledge efficiently. However, due to the limitation of our computation resources, we cannot perform our experiments on larger PTLMs such as T0 or GPT-4. However, we believe that while optimizing hyperparameters should yield higher performance, the trends between different pipelines should remain the same (i.e., knowledge injection should still be beneficial).

Additionally, although the scope of this paper is to understand the effect of idiomatic knowledge injection on IE comprehension, it would be more beneficial for the community to generalize our method to more domains that require domain knowledge, e.g., medical domain, legal domain, and science domain. However, our experiments are limited by focusing only on the idiom domain.

## Ethics Statement

The proposed system is intended to use as a method to enhance pre-trained language models' ability to interpret and comprehend the figurative semantics of idiomatic expressions and thus improve the performances on various natural language understanding tasks, especially those that need to pay close attention to the presence of idiomatic expressions in the text. In case of failure, the system will misinterpret the meaning and uses of idiomatic expressions, thus negatively impacting the performances of any related downstream task. Given that we have not thoroughly studied the causes of failures or the performances of idioms not covered by our idiomatic knowledge injection phase, we do not recommend the uses in

critical domains. During the training and testing of our methods and their downstream applications, we only use publicly available datasets without collecting or generating additional data that could breach privacy rights. Hence, when used properly, our work presents no calls for ethical concern.

## Acknowledgements

This research was supported by the National Science Foundation under Grant No. IIS 2230817 and in part by the U.S. National Science Foundation and Institute of Education Sciences under Grant No. 2229612.

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

# A Appendix

## A.1 Experimental Settings

Our experiments are performed as follows: we use the T5-large model Raffel et al. (2020) as the backbone PTLM. We add Idapter to the backbone model, which is then continued pretrained to perform the idiomatic knowledge injection using our multi-view data augmentation method. For this continued pre-training stage, we train the Idapter for 5 meta-epochs within each of which the model is trained for 50 epochs with a batch size of 32 using the Adam optimizer with a learning rate of 2e-5 and a linear decay scheduler with 60 warm-up steps. For outer learning rate of meta-learning is 1e-5. After injection, we deploy the model on different downstream tasks, where we use the parameters initialized from the trained Idapter. We train for 20 epochs with a batch size of 64 and the same optimizer and scheduler setup as before. We apply prompt templates shown in Table 1 to all the tasks during training and inference to convert each example into an instructive text-to-text format. Importantly, we apply this recipe to every downstream dataset in exactly the same way without modifications.

## A.2 Downstream Tasks

**IE-related Natural Language Inference.** This semantic understanding task involves determining whether a given hypothesis can be inferred from a given premise (entailment) or not (non-entailment). The overall challenge is compounded with the insertion of an IE in the Idiomatic and Metaphoric Paired Language Inference (IMPLI) dataset (Stowe et al., 2022). For this dataset, E represents entailment by paraphrasing idioms in the premise, NE represents non-entailment created from adversarial idiom definitions, and ANT represents non-entailment created by swapping keywords in idiom definitions to their antonyms.

**Idiomatic Expression Paraphrasing and Literal Expression Stylizing.** Proposed by (Zhou et al., 2021c), the tasks involve paraphrasing IEs into literal phrases and stylistically transforming the literal phrases in given sentences by replacing them with their IE counterparts, respectively. We use the dataset created by (Zhou et al., 2021c) for this task (denoted as I2L and L2I henceforth).

**Figurative Narrative Benchmark.** Proposed by (Chakrabarty et al., 2021a), the task is to continue a given narrative by choosing an acceptable continuation from two candidates. The dataset has narratives that use idioms and similes. Following the settings in (Zeng et al., 2023) and to account for our setting of working with idioms, we only utilize the narratives with idioms. Moreover, we transform each instance into two continuation acceptability examples: one that pairs the narrative with the correct continuation, and another that pairs the same narrative with the incorrect continuation (denoted as FNB henceforth).

## A.3 Performance on Compositional Tasks

Here we show that injecting idiomatic ability does not take away our model's prior ability with compositional expressions. We perform experiments on sentiment classification and paraphrase identification following the settings in Zeng & Bhat (2023). For sentiment classification, we use the SST2 (Socher et al., 2013) dataset and its default train and test splits (two classes). For paraphrase identification, we combine the MRPC (Dolan & Brockett, 2005) and PAWS (Zhang et al., 2019a) datasets and their default train/test splits with a total of 53,069 train and 9,725 test instances. The experimental results are presented in Table 8. As shown in Table 8, compared with the original T5-large, our model still achieves a comparable performance for these two tasks, which proves that injecting idiomatic ability does not take away our model's prior ability with compositional expressions.

## A.4 Performance on More Types of Non-compositional Expressions

Here we show that injecting idiomatic ability could also help with the comprehension of other types of non-compositional expressions including metaphors and euphemism. We

| Method | I2L | | | L2I | | |
|---|---|---|---|---|---|---|
| | BLEU | Rouge-2 | Bertscore | BLEU | Rouge-2 | Bertscore |
| Full | 82.46 | 77.15 | 81.32 | 82.89 | 76.69 | 80.46 |
| Adapter | 81.21 | 75.36 | 80.14 | 82.72 | 76.66 | 80.03 |
| Lora | 81.12 | 75.28 | 80.20 | 81.85 | 75.74 | 80.09 |
| $(IA)^3$ | 76.44 | 69.82 | 71.74 | 76.38 | 70.19 | 73.08 |
| Llama2-10shot | 61.09 | 56.36 | 78.97 | 65.38 | 60.63 | 80.22 |
| Mistral-10shot | 77.12 | 69.24 | 71.66 | 77.88 | 70.55 | 73.28 |
| Ours | **86.11** | **81.32** | **86.46** | **85.78** | **80.79** | **85.40** |

Table 6: Performance of different methods for i2l and l2i downstream tasks. Here we include the Bertscore for more thorough evaluation. When evaluating using Bertscore, we only extract out the idiomatic expressions and their corresponding literal parts. For I2L, we directly calculate the bertscore based on generated literal parts and reference literal parts. For L2I, we calculate the Bertscore between the generated idioms and the reference idioms.

| Dataset | Model | #Para | E | NE | ANT | Acc |
|---|---|---|---|---|---|---|
| **MNLI** | **Full** | 770M | 75.04 | 39.11 | 75.81 | 67.41 |
| **MNLI+PT** | **Ours** | 3M | 82.47 | 40.35 | 77.91 | 75.45 |

Table 7: Performance of zero setting for IMPLI task.

perform experiments on metaphor detection and euphemism detection. We use the VUAall dataset (Leong et al., 2018) and Euphemism dataset (Gavidia et al., 2022) for our experiments. The experimental results are presented in Table 9. As shown in Table 9, compared with the original T5-large, our model still achieves a better performance for these two tasks, which proves that injecting idiomatic ability could also help with the comprehension of metaphors and euphemism.

## A.5  Experiments on LLMs

Here we show the experimental details of Llama2-13b and Mistral-7b, including the prompts used and different performance when different numbers of in-context examples are used.

In the following we provide the prompts we used :

**IMPLI:**

Below is an instruction that describes a task. Write a response that appropriately completes the request.

### Instruction:

If the following premise and hypothsis are in entailment or contradiction:

Premise: [sentence]

Hypothesis: [sentence]

Just output the entailment or contradiction in the Response.

### Response: [entailment/contradiction]

**I2L:**

Below is an instruction that describes a task. Write a response that appropriately completes the request.

### Instruction:

| Method | Sentiment Classification | | | Paraphrase Identification | | |
|---|---|---|---|---|---|---|
| | Positive | Negative | Acc | Positive | Negative | Acc |
| Full T5-large | 96.17 | 96.03 | 96.10 | 94.11 | 92.30 | 93.17 |
| Ours | 96.85 | 95.56 | 96.22 | 93.98 | 92.32 | 93.12 |

Table 8: Performance of different methods for sentiment classification and paraphrase identification.

| Method | Metaphor Detection | | | Euphemism Detection | | |
|---|---|---|---|---|---|---|
| | Literal | Figurative | Acc | Literal | Figurative | Acc |
| Full T5-large | 95.29 | 77.22 | 92.73 | 83.82 | 89.06 | 87.24 |
| Ours | 96.47 | 75.80 | 93.53 | 86.76 | 89.06 | 88.27 |

Table 9: Performance of different methods for metaphor detection and euphemism detection.

Paraphrase the following sentence via only paraphrasing the idiom into its literal counterparts: [sentence]

### Response: [sentence]

**L2I:**

Below is an instruction that describes a task. Write a response that appropriately completes the request.

### Instruction:

Paraphrase the following sentence into its counterpart with an idiom: [sentence]

### Response: [sentence]

**FNB:**

Below is an instruction that describes a task. Write a response that appropriately completes the request.

### Instruction:

Given the following sentences: [sentence]

If the following continuation correct or not: [sentence]

Just output correct or wrong in the Response.

### Response: [correct/wrong]

| Method | IMPLI | | | | I2L | | L2I | | FNB | | |
|---|---|---|---|---|---|---|---|---|---|---|---|
| | E | NE | ANT | Acc | B | R | B | R | CC | ICC | Acc |
| Llama2-2shot | 53.32 | 56.17 | 73.91 | 68.88 | 54.96 | 49.53 | 62.04 | 57.80 | 71.27 | 66.76 | 69.01 |
| Llama2-6shot | 50.80 | 86.75 | 91.25 | 71.79 | 60.91 | 56.31 | 65.27 | 60.52 | 66.48 | 78.87 | 72.68 |
| Llama2-10shot | 53.32 | 84.19 | 89.84 | 71.92 | 61.09 | 56.36 | 65.38 | 60.63 | 62.82 | 81.41 | 72.11 |
| Mistral-2shot | 40.61 | 73.91 | 75.13 | 59.10 | 75.75 | 73.16 | 72.44 | 70.40 | 71.55 | 35.77 | 53.66 |
| Mistral-6shot | 30.74 | 90.51 | 91.44 | 63.52 | 76.73 | 74.06 | 73.61 | 70.36 | 76.34 | 56.90 | 66.62 |
| Mistral-10shot | 36.05 | 84.58 | 90.11 | 64.21 | 77.12 | 74.27 | 73.88 | 70.55 | 74.08 | 59.72 | 66.90 |

Table 10: Performance of Llama2-13b and Mistral-7b.

