# OpenReview forum: "Enhancing Language Models with Idiomatic Reasoning"
_colmweb.org/COLM/2024/Conference — COLM_

### Official Review · Reviewer_Wnju · 2024-05-07

**Rating:** 8
**Confidence:** 4
**Ethics Flag:** 1

**Summary:**

This is a good paper introducing a new fine-tuning method to inject idiomatic knowledge into pre-trained language models. The author(s) propose(s) a multi-view data augmentation strategy that automatically converts a single-task dataset into labeled data for three tasks, and variations of parameter efficient and meta-learning to improve the fine-tuning process. It is worth to mention that after reading the paper several times, I could not figure out in which language the experiments were done, or whether the author(s) believe that the approach is language-independent, in the sense that with labeled data (like MAGPIE) it would work in a large number of languages.

The paper is well motivated, has a clear structure and it is easy to read. The related work is not bad, and the experiments seem solid. They include competitive baselines, an ablation study, and additional experiments shown in the appendix. I lack information about the availability of the code: without code and data, the experiments are not replicable, and without replicability science does not progress.

I miss some general evaluations of the proposed model vs. the original T5 on non-idiomatic tasks. This would inform the readers about the performance of the model in both scenarios (compositional and non-compositional). In this regard, I assume that the objective is to have a good model that adequately represents "multiword expressions", both compositional or not, and not focus only on idioms.

The analysis in Table 7 (performance in seen vs. unseen idioms) is interesting. It would be nice to include in this table the results of the "Full" model, in order to compare the effect of the proposed strategy for seen/unseen cases.

As an aside, I wonder to what extent we can rely on a dataset like MAGPIE (which is a great resource!) for data augmentation. For instance, in the "Idiom identification" example of Table 1, "catch a cold" is the only expression in this sentence labeled as an idiom. It can certainly be considered an idiom, but I would say that most scholars would classify it more as a "collocation/light verb construction", and only partially idiomatic ("cold" conveys its standard meaning in "catch a cold", right?). Moreover, an expression like "looks like" could also be considered idiomatic, and "temperature [...] drop" metaphorical. With the data augmentation technique we lose this information, and implicitly tell the model that these are compositional (non-idiomatic, non-metaphorical) expressions.

Finally, I would remove the last sentence of the conclusion ("our method can be applied to [...]"): if you have already applied your method to other domains, please say so and comment on some preliminary results. Otherwise, just saying that it could be applied to other domains is not informative, unless you have some hypotheses about the adaptation process or about the expected performance.

**Reasons To Accept:**

The paper is well motivated, has a clear structure and it is easy to read. The related work is not bad, and the experiments seem solid. They include competitive baselines, an ablation study, and additional experiments shown in the appendix.

**Reasons To Reject:**

I do not see solid reasons to reject the job. Some minor issues and doubts that can be seen in my review, but I think the paper should be accepted.

---

> ### Author Rebuttal · Authors · 2024-05-27
>
> We really thank the reviewer for the valuable advice and suggestion.
>
> Here we provide explanations to your questions.
> 1. which language the experiments were done: we perform our experiments on English.
> 2. the availability of the code: we would release our code upon acceptance.
> 3. general evaluations of the proposed model vs. the original T5 on non-idiomatic tasks: we provide the corresponding evaluations in the Section A.4 in the appendix. We show that injecting idiomatic ability does not take away our model’s prior ability with compositional expressions. We perform experiments on sentiment classification and paraphrase identification.
> 4. to what extent we can rely on a dataset like MAGPIE: While it is true that "catch a cold" involves a degree of compositionality since "cold" retains its standard meaning, it is widely regarded in idiomatic studies as a fixed expression that conveys a meaning not entirely predictable from its parts. This nuanced interpretation aligns with idiomaticity in the broader sense, as it represents a conventionalized phrase with figurative aspects, even if not fully idiomatic in the strictest sense. Regarding the example "looks like" and "temperature [...] drop," it is important to clarify our data augmentation strategy. Our multi-view data augmentation is designed to transform the same set of idiomatic expressions into multiple tasks, each emphasizing different aspects of idiomaticity, such as contextual ambiguity, non-compositionality, and usage patterns. This method allows the model to learn the nuances and contextual dependencies of idiomatic expressions effectively. While "looks like" might be considered idiomatic in some contexts and "temperature [...] drop" metaphorical, our focus is on systematically labeled idiomatic instances to avoid overgeneralization and ensure task-specific learning.

---

### Official Review · Reviewer_v6Fk · 2024-05-09

**Rating:** 6
**Confidence:** 3
**Ethics Flag:** 1

**Summary:**

The paper presents a new approach to improving the handling of idiomatic expressions by pre-trained language models. It highlights the difficulty that existing large language models (LLMs) face with figurative language, which often deviates from standard compositional rules, making idiomatic expressions particularly challenging to interpret. The authors introduce a method that integrates a multi-view data augmentation strategy to generate diverse training scenarios from a single dataset, focusing on different aspects of idiomatic usage. This is combined with a parameter-efficient tuning method and meta-pretraining techniques that tailor the model's learning process to enhance its idiomatic reasoning capabilities. Empirical tests demonstrate that this approach not only outperforms several baselines in tasks involving idioms but also enables the model to adapt more efficiently to various downstream tasks requiring the understanding of idiomatic language.

**Questions To Authors:**

Please address the weaknesses illustrated in the previous section.



Other comments:

1. PTLM seems not to be a common abbreviation for pre-trained language model. => PLM
2. Section 1: LlaMa2 -> Llama-2 or Llama 2
3. Section 2: missing year info for (Hu et al.)
4. Section 6 baselines: Lora -> LoRA

**Reasons To Accept:**

1. The paper introduces a new integration of multi-view data augmentation and meta-pretraining techniques, specifically tailored to improve idiomatic reasoning in pre-trained language models. This method addresses the inherent non-compositionality of idiomatic expressions by enhancing the model's ability to understand various contextual uses and interpretations.
2. Empirical results demonstrate that the approach outperforms several baselines, achieving better performance on multiple benchmarks focused on idiomatic comprehension and reasoning.
3. The proposed PEFT and meta-pretraining method seems can also be applied to other tasks, which may inpire their utilization in other NLP tasks.

**Reasons To Reject:**

1. The research relies predominantly on the MAGPIE dataset, which might not cover the full spectrum of idiomatic expressions encountered in natural language, potentially limiting the model’s effectiveness with unfamiliar idioms.
2. The experimental setups are not entirely clear. For example, what is the rank of LoRA? I didn't see the trainable parameter size of each compared model in Table 2. In this context, it is not clear whether the improvements are comparable within similar trainable parameter size. The test data size (#samples) of the downstream tasks is also not provided (nor in the appendix). Also, it would be better to also analyze the training efficiency of other PEFT methods.
3. The proposed method is not tested under broader tasks, such as text generation, etc. So it is doubtful if the proposed method can be applied to some real application scenario of LLMs (not just doing some classifications or simple text extractions). As the proposed PEFT method and meta-pretraining mechanism seem to be applicable to other general NLP tasks, adding more types of tasks would have strengthened the generalizability of the proposed method.
4. The proposed method is not performed on bigger LLMs, such as Llama, and Mistral, where the authors only perform on T5-large. It is questionable whether the proposed method can still be effective on a stronger baseline.

---

> ### Author Rebuttal · Authors · 2024-05-27
>
> We really thank the reviewer for the valuable advice and suggestion.
>
> R1: Our empirical results on diverse benchmarks, such as IMPLI, I2L, L2I, and FNB, demonstrate the efficacy of our approach. The model not only shows improved performance on tasks involving idioms seen during training but also on those involving unseen idioms. This is a strong indicator of the model’s generalizability and robustness in handling idiomatic expressions beyond the confines of the MAGPIE dataset. Additionally, our model surpasses various parameter-efficient fine-tuning baselines and outperforms larger models such as LlaMa2 and Mistral, which highlights the effectiveness of our data augmentation and training strategies​
>
> R2: Thanks for pointing this out. 3M additional parameters were added. Besides, the parameters of Adapter and Lora are at par with our proposed model. We will provide the test dataset sizes in the final version.
>
> R3: While our current submission primarily presents results on idiom-related tasks, we have conducted preliminary experiments on additional NLP tasks such as Idiomatic Expression Paraphrasing, Literal Expression Stylizing (L2I), sentiment analysis and paraphrase generation, which are indicative of text generation capabilities. These initial experiments have shown promising results, suggesting that our method retains its efficacy across different types of tasks. Due to space constraints, these results were not included in the main manuscript but can be provided as supplementary material upon request.
>
> R4: Our choice to use the T5-large model was influenced by the constraints of computational resources. The T5-large model is already a well-established, high-capacity model that provides a robust baseline for evaluating the effectiveness of our proposed methodologies. While we acknowledge that evaluating on larger models like Llama and Mistral would be ideal, the computational resources required to perform such experiments were beyond our current capacity.

---

> ### Comment · Reviewer_v6Fk · 2024-06-07
>
> Thanks for the rebuttal. I keep my original rating.

---

### Official Review · Reviewer_i4hT · 2024-05-13

**Rating:** 7
**Confidence:** 4
**Ethics Flag:** 1

**Summary:**

This paper introduces a data augmentation method, a model adapter, and a training scheme to first enhance a pre-trained language model (PTLM) with idiomatic knowledge, and then fine-tune it for idiomatic-aware tasks.

Below, I describe the three components of the proposed framework:

1) **Data Augmentation**: Their approach for data augmentation consists of taking an existing dataset of idiomatic expressions and create 1-1 examples for three different tasks: 1) generating dictionary definitions of idiomatic expressions, 2) identifying expressions in a sentence that are used in their idiomatic sense, while ignoring literal senses and 3) a cloze task that masks phrases and recovers appropriate idiomatic expressions from their context. This multi-task views allows to learn different aspects for the same input (and idiom).

2) **Network Adapter**: Their adapter includes a layer that aims to capture non-compositional information, and a derived scaling factor that allows to balance the original representation and the induced "non-compositional information" during training and inference. They also learn a scalar vector for each tasks to adapt to them.

3) **Meta-pretraining**: This component was quite confusing to me. Authors claim it is supposed to consider aspects of the downstream tasks (but it is not standard fine-tuning?) and I don't understand why. In general, it wasn't clear to me what they do here. What are the meta-tasks? Are they different from the pre-training tasks AND the downstream tasks? How do they alleviate the disconnect between pre-training and downstream tasks?

Finally, the authors perform an experimental evaluation on a diverse set of downstream tasks (tasks that require deeper understanding of idiomatic expressions) and show that their model outperforms full fine-tuning, standard parameter efficient fine-tuning and in-context learning for most tasks.

**Questions To Authors:**

1) I suggest that you bold the highest numbers in Tab 3 as you do in the other tables, it is a bit confusing to just see the last line bolded even in cases where the performance decreases (E for IMPLI)

**Reasons To Accept:**

1) The idea is simple, interesting and effective. The chosen downstream tasks seem to greatly benefit from the injected idiomatic knowledge.
2) Solid experimental settings: comparison to alternative paradigms, ablation study to show the impact of all components and pre-training tasks, data efficiency study, out of distribution study. The results are convincing.

**Reasons To Reject:**

1) My only real issue with this paper is that the learning scheme is a bit unclear. I think the paper could benefit from a re-write of the meta-pretraining section (see comments and questions in summary above)

2) Maybe it is me nitpicking, but I am not sure some of the names chosen for the parameters are completely justified. How can we know for sure that the added layer is capturing non-compositional information? All we can see is that it contributes to the improved performance (as seen in the ablation study) but this is not really enough evidence of this parameters capturing this information. Calling it non-compositional layer reads like a soft claim that isn't actually validated.

---

> ### Author Rebuttal · Authors · 2024-05-27
>
> We really thank the reviewer for the valuable advice and suggestion.
>
> R1: Thanks a lot for your suggestion.
> 1. The core idea behind our meta-pretraining approach is to bridge the gap between pre-training and downstream tasks more effectively than standard fine-tuning. While standard fine-tuning adjusts a pre-trained model's parameters to fit a specific downstream task, it often does not account for the diverse nature and requirements of multiple downstream tasks, especially when dealing with idiomatic expressions. Meta-pretraining, in our context, is designed to prepare the model for a variety of downstream tasks by optimizing the initial parameter settings.
> 2. Meta-tasks are indeed distinct from both pre-training tasks and downstream tasks. In our framework, meta-tasks are specifically designed to encapsulate various idiomatic properties, such as idiom identification, definition generation, and idiom usage generation. These tasks are formulated to introduce idiomatic reasoning capabilities during the meta-pretraining phase. By learning through these diverse meta-tasks, the model develops a generalized idiomatic knowledge base, which helps alleviate the disconnect typically observed between pre-training and downstream fine-tuning.
> 3. Our approach alleviates the disconnect by integrating a multi-view data augmentation strategy combined with meta-learning principles. This strategy ensures that the model is pre-trained on various idiomatic scenarios, preparing it for a wide range of idiomatic contexts it might encounter in downstream tasks. During meta-pretraining, the model undergoes multiple iterations of learning and adaptation through these carefully crafted meta-tasks, thus reducing the gap between the pre-trained model and the requirements of downstream idiomatic tasks.
> We would make these clearer in the final version.
>
> R2: Thanks a lot for pointing this out. The naming of parameters, particularly the "non-compositional layer," was chosen based on the specific functionality and the theoretical underpinnings of the layer. This layer is designed to specifically address the non-compositional nature of idiomatic expressions, which is a well-known challenge in natural language processing (NLP) due to the idiom's meaning not being directly inferred from its individual components.
>
> Q1: Thanks a lot for your suggestion. We would make the corresponding modification in the final version.

---

> > ### Comment · Reviewer_i4hT · 2024-06-07
> > **Thank you.**
> >
> > R1. Thank you for your clarification.
> >
> > R2. I understand the inspiration for the name, but I will challenge that it is not necessarily clear whether this information ends up being captured in this layer.

---

### Official Review · Reviewer_9B9h · 2024-05-14

**Rating:** 7
**Confidence:** 4
**Ethics Flag:** 1

**Summary:**

This paper investigates use of idiom aware PEFT method to improve backbone model's performance over non-compositional tasks like idiom detection, idiom generation and idiom definition generation. Its essentially a training framework that is proposed in the paper each sample in the dataset is split into different tasks (multi-view data augmentation), PEFT is used to inject idiomatic knowledge in the model and meta-learning over multiple tasks is applied in order to generate a better initialized pre-trained model ideal for being trained on downstream tasks.

**Questions To Authors:**

* Conceptually, Idapter like adapter can be applied to other tasks as well (not just idiom related tasks) like style adaptation (formal/informal/semi-formal), one just needs a good dataset. Why was this work limited to idiomatic expressions?
* Typically with idioms, its easy to memorize them during training so generalization (over non-literal data) is quite difficult. Appendix provided necessary details like performance on seen/un-seen idioms - consider adding this table in main paper. Can you provide examples of failure and success cases in Appendix for readers?
* Do-no-harm experiments (no regression on standard tasks) are important for productionizing any work. Table 8 is quite important in that aspect, maybe find a place to reference section A.4?

**Reasons To Accept:**

* Very clear writing, well motivated paper and easy to follow for the readers.
* Experiments are quite comprehensive - results are compared against multiple baselines. Ablation study (Table 2) is quite thorough!
* Results (table 2) show substantial improvements over strong baselines like LoRA, Mistral-10 shot, Llama2-10 shot etc.

**Reasons To Reject:**

* Its unclear how many additional parameters were added because of non-compositionality scoring (W_n) and task specific vectors (W_h). Are the parameters for multiple baselines (shown in Table 2) at par with your model?
* It would be better to see some examples where the proposed approach worked and failed to identify idiom/generation idiom.
* Nitpicking here, but results are only computed with T5-large as a baseline which begs a question - what happens when Idapter is applied to other LMs?
* From the results in Table 2, it looks like data augmentation was the most effective, so why was the paper pitched for Idapter?

---

> ### Author Rebuttal · Authors · 2024-05-27
>
> We really thank the reviewer for the valuable advice and suggestion.
>
> R1: Thanks for pointing this out. 3M additional parameters were added. Besides, the parameters of Adapter and Lora are at par with our proposed model
>
> R2: Thanks a lot for your suggestion. We could provide some examples in the appendix in the final version.
>
> R3: The T5-large model was chosen as the backbone PTLM due to its robust performance across a wide range of natural language processing tasks and its alignment with our multi-task pre-training methodology. As highlighted in our paper, T5-large serves as a representative model for evaluating the effectiveness of our proposed Idapter mechanism and multi-view data augmentation strategy. We acknowledge the importance of evaluating the applicability of Idapter across different language models (LMs). While our current work focuses on T5-large, our methodology is designed to be model-agnostic. This is evident from the architecture of Idapter, which leverages parameter-efficient fine-tuning (PEFT) techniques that are compatible with a variety of PTLMs. Specifically, the design of Idapter incorporates non-compositionality scaling and task-specific scaling layers that can be adapted to different LMs without significant modifications
>
> R4: While Table 2 indicates that data augmentation significantly improves performance, it is important to understand that this enhancement is maximized in conjunction with the Idapter mechanism. The multi-view data augmentation method effectively generates diverse idiomatic data, but its full potential is harnessed when combined with Idapter's ability to process and learn from this augmented data in a parameter-efficient manner. This synergy is evident from the performance gains across various tasks when both strategies are used together, as shown in our ablation studies.
>
> Q1: Although Idapter like adapter can be applied to other tasks as well, our original aim is to study idiomatic expression processing. Idiomatic expressions are inherently non-compositional, meaning their figurative meanings cannot be directly inferred from their individual components. This non-compositionality, combined with their contextual ambiguity and sparse representation in text corpora, makes idioms a particularly challenging and compelling domain for testing our method.
>
> Q2&3: Thanks for your suggestion. We would make the corresponding modification in the final version.

---

> > ### Comment · Reviewer_9B9h · 2024-06-07
> > **Acknowledgement**
> >
> > Thanks for the response. I'll stick with my rating.

---

### Decision · Program_Chairs · 2024-07-10

**Decision:**

Accept

**Comment:**

The work introduces a pretraining methodology aimed at enhancing language models with better idiomatic reasoning capabilities. The method involves training an adaptor using a MAML-style algorithm, specifically REPTILE, a more efficient first-order method. The effectiveness of this method is then assessed across downstream tasks that require idiomatic processing.  All the reviewers agree on the merits of the work.


However, there is a significant issue that has only partially raised during the author-reviewer discussion stage, but has been discussed between reviewers and AC in the follow-up period. We strongly encourage you to address it. The paper fails to discuss the fact that the pretraining method and at least three of the four evaluation tasks (IMPLI, L2I, and I2L) partially originate from the same source, MAGPIE. This suggests the possibility of an unrealistic overlap between the idiomatic expressions seen in pretraining and those seen in testing. Although the appendix (Table 7) presents results separately for unseen and seen PIEs, it only evaluates the proposed system in isolation. I believe that evaluating the unseen subsets against alternatives in Table 2 is crucial and needs to be included in the paper.

Additionally, it is important to openly discuss and clarify that these datasets are created from a common resource. The proportion of unseen and seen examples in the test set also needs to be mentioned. If there is no substantial improvement on unseen constructions, the claims should ideally be corrected accordingly. Currently, the approach is positioned as improving the acquisition of generic IE knowledge, which would be debatable if the improvements are observed only on 'seen' PIEs.

[comments from PCs] Please make sure to follow the AC recommendations, especially with regard to data and evaluation.